# Systematic Investigation of Plant-Parasitic Nematodes Associated with Main Subtropical Crops in Guangxi Province, China

**DOI:** 10.3390/life11111177

**Published:** 2021-11-03

**Authors:** Yi-Xue Mo, Ai-Su Mo, Zhuo-Qiu Qiu, Bing-Xue Li, Hai-Yan Wu

**Affiliations:** Guangxi Key Laboratory of Agric-Environment and Agric-Products Safety, Agricultural College of Guangxi University, Nanning 530004, China; 2017304017@st.gxu.edu.cn (Y.-X.M.); winter500@126.com (A.-S.M.); athos0519@163.com (Z.-Q.Q.); 18437958381@163.com (B.-X.L.)

**Keywords:** plant parasitic nematodes, rice, maize, sugarcane, soybean, geographical distribution

## Abstract

Plant parasitic nematodes (PPNs) are a pathogenic group that causes momentous crop yield loss by retarding plant growth and development through plant parasitization. In this study, the distribution of PPNs based on the main crops in Guangxi Province of China was investigated. A total of 425 samples of soil or roots from sugarcane, rice, maize, and soybean were collected in 68 counties, and a total of 48 order/family/genera of PPNs were identified, of which some genera were found in more than one crop. A total of 31 order/family/genera of PPNs were found in rice, among which *Hirschmanniella* was the most abundant, accounting for 79.23%, followed by *Tylenchorhynchus* (34.43%). Forty order/family/genera were observed in maize, of which the dominant genera were *Pratylenchus* and *Tylenchorhynchus* at 45.14% and 32.64%, respectively. In addition, 30 order/family/genera of PPNs were detected from sugarcane, and the percentages of *Tylenchorhynchus* and *Helicotylenchus* were 70.42% and 39.44%, respectively. The main crop of Eastern ecological regions was rice, with a high frequency of *Hirschmanniella*. The greatest frequency of *Pratylenchus* was found in the Western eco-region, which had a large area of maize. In the Northern eco-region, rice and maize were popular, with abundant *Hirschmanniella* and *Helicotylenchus.* In the Central eco-region, *Pratylenchus* was detected on the main crop of sugarcane. *Hirschmanniella* (72.94%) was dominant in clay, and *Tylenchorhynchus* (54.17%) showed the highest frequency in loam. The distribution of PPNs varied with different altitudes. The diversity of this phenomenon was closely related to host plants. These results could improve understanding of the distribution of PPNs and provide important information for controlling PPNs.

## 1. Introduction

Plant parasitic nematodes (PPNs) are a serious menace to a variety of crop plants worldwide. More than 4000 PPNs have been reported, resulting in over USD 173 billion of economic loss annually [1,2]. The plants infected by nematodes experience dwarfism, discoloration, and blocked absorption of nutrients, thereby seriously affecting the yield and quality of agricultural products [3,4]. Multiple PPNs have been implicated in disease complexes with plant pathogenic fungi and bacteria; for instance, soybean cyst nematode can aggravate sudden death syndrome (*Fusarium virguliforme*) in soybean [5,6].

The species and distribution of PPNs are closely related to different crops. Rice could be attacked by more than 200 plant pathogenic nematodes, and rice parasitic nematode disease has become increasingly serious with the promotion of water-saving cultivation mode [7], causing an annual loss of approximately USD 16 billion worldwide [8]. The main parasitic nematodes of rice include *Meloidogyne* spp., *Aphelenchoides* spp., *Hirschmanniella* spp., *Ditylenchus* spp., and *Heterodera* spp. [9]. *M. graminicola* can cause substantial yield losses, representing up to 87% of production [10]. In Guangxi, *Heterodera elachista* and *Meloidogyne*
*graminicola* have been reported on rice [11,12]. At least 120 PPNs can parasitize maize worldwide, including more than 60 species in North America, and *Pratylenchus* spp. and *Helicotylenchus* spp. can cause immense harm to maize. Maize yield losses of 50% have been reported in Brazil, where lesion nematodes are present [13]. Several studies have uncovered PPNs diversity in sugarcane fields in countries such as Mauritius, India, Kenya, South Africa, Brazil, and Japan, revealing the most common PPNs genera associated with sugarcane as *Meloidogyne*, *Pratylenchus*, *Tylenchorhynchus*, *Rotylenchulus*, and *Helicotylenchus* [14]. In China, *Tylenchorhychus* spp., *Helicotylenchus* spp., *Meloidogyne* spp., and *Pratylenchus* spp. seriously damage sugarcane [15]. South Africa has reported 10–30% yield losses caused by PPNs, and greater than 40% of occasional yield reduction caused by PPNs was reported in Africa [16,17]. Soybean is also damaged by nematodes, such as *Heterodera glycines*, *Meloidogyne incognita*, and *Rotylenchulus reniformis*. Reports have pointed out that *H. glycines* can cause billions of dollars of economic losses worldwide and 30–50% of production reduction in Northeast China and Huanghuai soybean production areas [18,19,20].

The distribution of nematodes is also related to environmental factors, such as soil type, rainfall, and altitude. Studies have indicated that soil type could affect the abundance, diversity, distribution, and pathogenicity of nematodes [9,21]. A previous study has pointed out that nematodes have fewer species, but a higher density in clay and loam than in sand. The concentrations of *Helicotylenchus* spp. and *Basiria* spp. are relatively lower in sand than in clay and loam [22]. As for studies of altitude, nematode diversity was affected neither by altitude nor by layer. However, the community structure changed with altitude [23], whereas studies have pointed out that nematodes are mainly linear with altitude (low nematode abundance at high altitude) [24].

Guangxi Province is located in the west of southern China, covering an area of 236,700 km^2^. Rice, maize, sugarcane, and soybean are the main food and cash crops. The geographical distribution and groups of PPNs provide the basis for the control of nematode diseases. However, a systematic investigation has not been carried out, and the parasitic nematode on these crops remains unexplored. This study aimed to (1) identify to the genus level PPNs from the rhizosphere soil of main crops, (2) reveal the distribution of PPNs in different ecological areas and crops in Guangxi, and (3) understand the distribution of PPNs in different soil types and altitudes.

## 2. Materials and Methods

### 2.1. Collection of Soil Samples

Soil was collected in 2016–2017 from the rhizosphere of rice, maize, sugarcane, and soybean in 68 counties of 14 cities in Guangxi Province. One centimeter of topsoil was removed before soil sample collection at a 15 cm depth; 5–10 cores around plants were collected with a soil auger in a Z-pattern in each plot [25]. The soil sample with some roots was placed in separate Ziploc bags and labeled with site, longitude, latitude, and altitude. They were then transported to the lab and stored at 4 °C for further analysis within 3 days.

### 2.2. Extraction of Nematodes

Roots from soil samples were collected by hand, the soil was gently mixed well, and the nematodes were extracted from a 100 mL soil sample using Cobb’s sieving and decanting techniques through 40-, 100-, and 325-mesh sieves. The sediments on the 100-mesh sieve were carefully transferred on a 5-cm diameter petri dish, and the cyst was examined under a stereoscopic microscope (Olympus SZX2-ILLT, Olympus Corporation, Tokyo, Japan). Nematodes were isolated from the collected sediments (on the 325-mesh sieve) using modified Baermann’s funnel [26]. Roots were cleaned completely and cut into segments 0.5-cm long to extract any nematodes using same method above. The nematodes were collected after 10–12 h; all extracted nematodes in each sample were identified using an optical microscope according to order, family, and genus (if possible) based on morphology [27]. The percent of genus nematode in all samples was calculated and expressed as a detection percentage (%) = (number of samples with a genus/total number of samples) × 100.

### 2.3. Soil Texture

Soil texture was classified in accordance with the international classification standard of soil texture and divided into clay, loam, and sand.

### 2.4. Ecological Regions in Guangxi Province

Guangxi is located in the Southern frontier region of China at 21°38′–26°23′ N and 104°29′–112°04′ E, which has a long sunshine time, high annual average temperature, and annual rainfall that vary greatly in regions. In accordance with the principle of climate similarity and the law of regional differentiation, taking the direction of agricultural development as the guidance, we comprehensively considered various factors for regional division [28,29]. Five ecological areas were divided into the Eastern Guangxi eco-region, Western Guangxi eco-region, Southern Guangxi eco-region, Northern Guangxi eco-region, and Central Guangxi eco-region.

### 2.5. Figures

All figures were created using SigmaPlot 12.5 and Photoshop 6.0.

## 3. Results

### 3.1. Identification of PPNs at the Genus Level

A total of 425 soil samples (183, 144, 71, and 27 samples from rice, maize, sugarcane, and soybean, respectively) were collected in fields located in the five ecological regions of Guangxi, and 48 order/family/genera of PPNs were identified (Table 1). The high-frequency genera included *Tylenchorhynchus*, *Pratylenchus*, *Helicotylenchus*, *Meloidogyne*, and *Hirschmanniella* in samples from rice, maize, sugarcane, or soybean.

In rice fields, 31 order/family/genera were observed, among which *Hirschmanniella* was the most abundant at 79.23%, followed by *Tylenchorhynchus* at 34.43%, and *Filenchus* and *Meloidogyne* at 26.78%. The other genera, such as *Pratylenchus* and *Helicotylenchus*, occurred sporadically at less than 10%.

Forty order/family/genera were detected in the soil samples from maize fields. *Pratylenchus* was associated with 45.14% of samples, followed by *Tylenchorhynchus* with 32.64%, *Helicotylenchus* with 26.39%, and *Filenchus* with 22.22%. *Heterodera zeae* was only found in the maize field of Xiaopingyang Town, Laibin, and was first reported in China [30].

Thirty order/family/genera of PPNs were isolated from the rhizosphere soil of sugarcane. *Tylenchorhynchus* exhibited the highest frequency at 70.42%, followed by *Helicotylenchus* at 39.44%. *Pratylenchus* showed 35.21% frequency, whereas *Meloidogyne* was not found in sugarcane.

Twenty-four order/family/genera from rhizosphere soil samples of soybean were found. *Helicotylenchus* was the most dominant, accounting for 33.33%. The frequency of *Tylenchorhynchus* was 29.63%, whereas *Pratylenchus* and *Filenchus* presented 25.93% frequency, and the proportions of *Aglenchus* and *Meloidogyne* were both 11.11%.

*Tylenchorhynchus* was found on the studied crops, which was arranged in the order from high to low frequency: sugarcane (70.42%) > rice (34.43%) > maize (32.64%) > soybean (29.63%). The frequency order of *Hirschmanniella* was as follows: rice (79.23%) > maize (9.03%) > sugarcane (7.04%) > soybean (3.70%). The order of *Helicotylenchus* on crops was as follows: sugarcane (39.44%) > soybean (33.33%) > maize (26.39%) > rice (4.38%). *Pratylenchus* displayed a high population, and the ratios were arranged in the following order: maize (45.14%) > sugarcane (35.12%) > soybean (25.93%) > rice (3.28%). *Meloidogyne* was not detected in sugarcane, and its ratio in rice, soybean, and maize was 26.78%, 11.11%, and 3.47%, respectively.

### 3.2. Occurrence of PPNs in Five Ecological Regions in Guangxi

The main crop in the Eastern eco-region is rice. The ratio of PPNs in this region was in the order of *Tylenchorhynchus* > *Hirschmanniella* > *Pratylenchus* > *Meloidogyne* > *Helicotylenchus*. Maize is the main crop in Baise, which belongs to the Western eco-region. The proportion of PPNs in this region was *Helicotylenchus* > *Pratylenchus* > *Hirschmanniella* > *Meloidogyne* > *Tylenchorhynchus*. The Southern eco-region includes Qinzhou, Fangchenggang, and Beihai. Rice and sugarcane are planted more in this region than any other crops. Guilin, Liuzhou, and Hechi belong to the Northern eco-region, and the principal crops are rice and maize. *Tylenchorhynchus* was dominant, followed by *Hirschmanniella*, *Helicotylenchus*, *Pratylenchus*, and *Meloidogyne*. In the Central eco-region, including Chongzuo, Nanning, and Laibin, the main crop was sugarcane, in which *Tylenchorhynchus* presented a high frequency. In contrast, *Hirschmanniella*, *Pratylenchus*, *Helicotylenchus*, and *Meloidogyne* displayed low population densities (Figure 1, Appendix A).

### 3.3. Main PPNs in Different Soil Textures

The results indicated that the distribution of PPNs was distinguished in different soil textures. In loam texture, the frequency sequence of the sampled crops was as follows: maize (51.39%) > sugarcane (26.39%) > rice (16.67%) > soybean (5.55%). *Tylenchorhynchus* displayed a high frequency, detected in 54.17%, and *Pratylenchus* accounted for 50% of the soil samples. *Helicotylenchus* and *Hirschmanniella* accounted for 31.94% and 26.39% of the analyzed samples, respectively, whereas *Meloidogyne* accounted for 2.78%.

In clay, the main planted crop was rice (90.59%), with a small amount of maize and sugarcane at 4.71% and 4.70%, respectively. The percentage of nematodes was in the order of *Hirschmanniella* (72.94%) > *Tylenchorhynchus* (38.82%) > *Meloidogyne* (21.18%) > *Helicotylenchus* (10.59%) > *Pratylenchus* (5.88%, Table 2, Figure 2).

### 3.4. PPNs in Regions with Different Altitudes

The distribution of PPNs varies at different altitudes. In contrast, some nematodes were distributed at all altitudes (Table 3), which were *Tylenchorhynchus, Hirschmanniella,*
*Meloidogyne, Pratylenchus, Helicotylenchus, Filenchus, Aglenchus, Pararotylenchus, Rotylenchus, Aphelenchus, Lelenchus, Helicotylenchus crenacauda, Bolendorus, Paraphelenchus, Coslenchus, Aphelenchoides, Ditylenchus*, and Dorylaimoidea respectively. At <50 m altitude, 25 order/family/genera were detected, with the frequencies of 42.55% and 45.68% for *Hirschmanniella* and *Tylenchorhynchus*, respectively. At 50–100 m altitude, 40 nematodes genera were found; the highest percentage was *Tylenchorhynchus* (42.86%). There were 28 order/family/genera at 100–150 m altitude, of which the most detected was *Hirschmanniella* with the frequency of 42.15%. *Hirschmanniella* and *Tylenchorhynchus* exhibited the same frequencies of 41.79% at 150–200 m altitude in 31 order/family/genera. When the altitude was >200 m among 30 kinds of PPNs, the nematode with the highest frequency was *Hirschmanniella* with 46.67%.

## 4. Discussion

In the present study, the PPNs in five ecological regions of Guangxi were investigated. On the basis of the morphological characteristics, 48 genera of PPNs were observed, and the important PPNs, including cyst nematode on maize and root knot nematode on rice, were identified as *Heterodera zeae* and *M. graminicola* [12,30]. The distribution of nematodes varied in different hosts, ecological regions, soil textures, and altitudes.

The large-scale occurrence of nematodes is closely related to continuous cropping, and the disease caused by nematodes is serious in the soil where nematodes are abundant [31]. In Ecuador, *Hirschmanniella* was frequently found in irrigated rice, whereas *Pratylenchus* had the greatest frequency in rainfed lowland rice; *Helicotylenchus*, *Criconemoides*, and *Tylenchorhynchus* were also abundant in rice fields [32]. In the present study, *Hirschmanniella*, *Meloidogyne*, *Tylenchorhynchus*, and *Filenchus* were dominant in rice and the detection frequency of *Hirschmanniella* was 79.23%, and this nematode was distributed in all rice areas in Guangxi; nine species of *Hirschmanniella* were identified and described in Guangdong Province. They are *Hirschmanniella areolata*, *H. behningi*, *H. anchoryzae*, *H. caudacrea*, *H. diversa*, *H. Meloidogyne* spp. *Pratylenchus* spp. *Helicotylenchus* spp. *Hirschmanniella* spp. *Tylenchorhynchus* spp. *orycrena*, *H. oryzae*, *H. microtyla,* and *H. mucronata* [33]. Thus, this is a great potential threat to rice-growing areas. It also occurred in a large area in Fujian and Jiangsu Provinces in China [34].

Maize had the most abundant parasitic nematodes in this survey. The dominant nematodes were *Pratylenchus* and *Tylenchorhynchus*, and the result was similar to the previous investigation on maize in Hebei, Shanxi, Henan, Shandong, and Jiangsu [35]. *Pratylenchus* is widely distributed, and a total of 100 species of *Pratylenchus* have been recorded worldwide [36]. The largest number of parasitic nematode genera in maize was identified in the present study, most of which had been previously reported. *Heterodera zeae* was first observed in China [30], which was considered a quarantined organism in the United States [37]. Previous studies on maize parasitic nematodes have primarily concentrated in Central and Northern China, and such nematodes have not been well explored in the Southern China. The present study provided a reference to parasitic nematodes in maize in Southern China. The results showed that the main parasitic nematodes were similar to North China, except *Heterodera zeae* was first reported on maize.

Over 310 species belonging to 48 genera have been reported from roots or rhizosphere of sugarcane, and *Meloidogyne*, *Pratylenchus*, and *Xiphinema* are the most economically damaging nematodes found in sugarcane [38]. Thirty genera of parasitic nematodes on sugarcane were observed in this study. *Tylenchorhynchus*, *Helicotylenchus*, and *Pratylenchus* were the dominant populations. *P. zeae* was frequently regarded as highly pathogenic to sugarcane [38,39].

*Heterodera*, *Meloidogyne*, or *Pratylenchus* cause severe damage in the main soybean-producing regions worldwide [40]. However, in this study, PPNs from soybean soil samples were *Tylenchorhynchus* and *Helicotylenchus*, which were widely distributed in Guangxi. In contrast, in Germany *Pratylenchus* was widely spread in soybean fields followed by *Rotylenchus* and *Paratylenchus* [41], probably because of the different geographical conditions and crop factors. In the present study, the small number of samples from soybean possibly resulted in the few nematode genera detected. Most studies on soybean parasitic nematodes have focused on *H. glycines* [42], which was not observed in this investigation.

*Tylenchorhynchus*, *Hirschmanniella*, *Helicotylenchus*, *Pratylenchus*, and *Meloidogyne* were the main genus observed in the five ecological regions. Rice was cultivated profusely in Eastern and Southern eco-regions; thus, a large proportion of *Hirschmanniella* was detected in the present research. Nematodes were associated with rice, wheat, maize, and legumes in Bangladesh, Nepal, and Pakistan [43]. Maize was the main crop in the Western eco-region, where a high frequency of *Pratylenchus* was presented. Rice and maize were common crops in the Northern eco-region, where *Hirschmanniella* and *Helicotylenchus* were observed the most. Sugarcane was primarily cultivated in the Central eco-region, where a high frequency of *Pratylenchus* was detected. These results suggest that the distribution of nematodes in different ecological regions was closely related to the main crop in the corresponding region.

Although the growth and pathogenicity of PPNs are influenced by soil conditions, including soil texture, moisture, and aeration in field soils [44], the compatibility between plant nematodes and their hosts is highly important. Therefore, rice in Guangxi is grown in clay soil, and *Hirschmanniella* can be found most frequently in clay.

Tsiafouli et al. [45] recently concluded that, among the factors that affect the nematode community, altitude was the most predictable because altitudinal climatic conditions strongly constrain the availability and turnover of basal resources, leading to the difference of nematode occurrence. Simultaneously, host plants affected the occurrence of nematodes in our study.

The present study pointed out that the PPNs were primarily distributed at 50–200 m, whereas fewer genera were distributed at <50 or >200 m altitude. The distinction among the results may be due to the crops being cultivated at different altitudes.

On account of samples, Dorylaimoidea nematodes were not identified at the genus level. The importance of this nematode group lies not only in their polyphagy and wide distribution but also in their status as vectors of plant viruses, which cause significant damage to a wide range of agricultural crops [46]. The dagger nematode *Xiphinema index* caused severe damage to grapevines, an essential crop in most Mediterranean countries; apart from the direct damage to the root, resulting in the typical stubby root with terminal swellings, *X. index* serves as the vector for grapevine fanleaf virus [47,48]. In addition, *Filenchus* has a high frequency in the present study; however, its importance as a plant pathogen is unknown [27].

## 5. Conclusions

Forty-eight order/family/genera of PPNs were observed on main subtropical crops in Guangxi province in this study, among which five had the highest frequency, including *Tylenchorhynchus*, *Pratylenchus*, *Helicotylenchus*, *Meloidogyne*, and *Hirschmanniella.* It was the host plants rather than soil texture and altitude that was closely related to the occurrence of PPNs. The results highlighted the need to monitor the population of *Hirschmanniella* in rice, *Pratylenchus* in maize, *Tylenchorhynchus* in sugarcane, and *Helicotylenchus* in soybean to protect crops. Effective control measures should be taken to control the nematode population in the future on the basis of nematode distribution on crops in ecological regions.

## Figures and Tables

**Figure 1 life-11-01177-f001:**
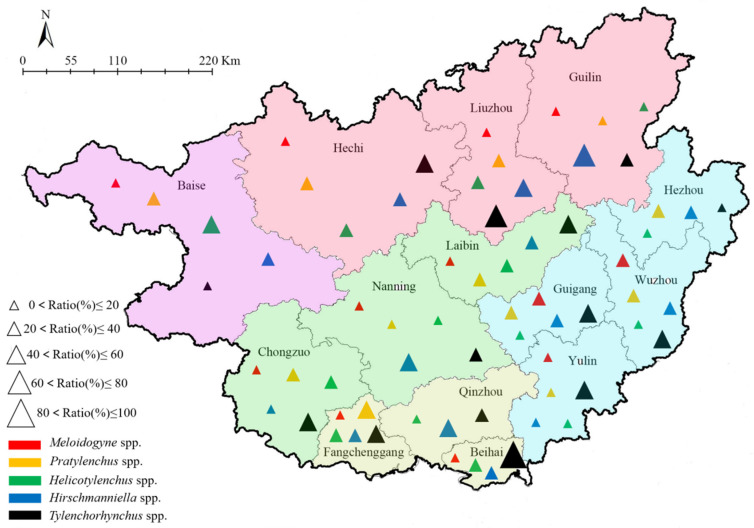
Distribution and proportion of major plant parasitic nematodes in different regions and cities.

**Figure 2 life-11-01177-f002:**
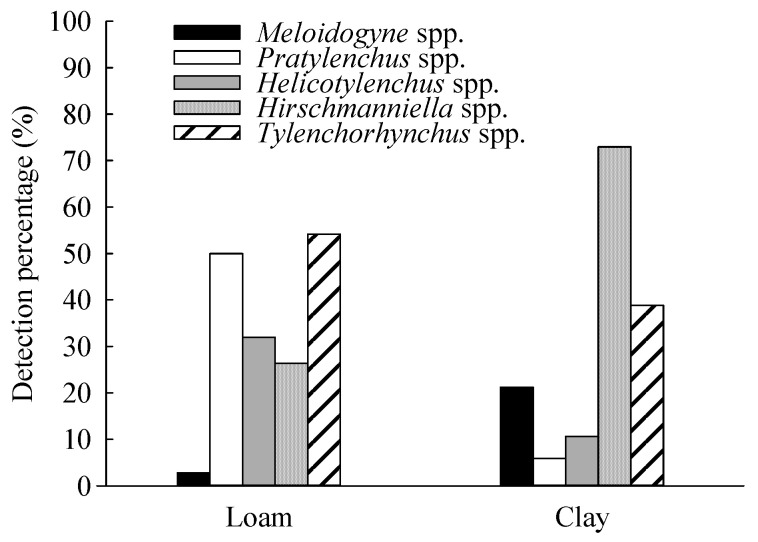
Detection percentage of plant parasitic nematodes in loam and clay soil.

**Table 1 life-11-01177-t001:** Genera and frequency ratio of plant parasitic nematodes detected in soil samples.

Order/Family/Genus	Rice		Maize		Sugarcane		Soybean	
	Samples	Ratio (%)	Samples	Ratio (%)	Samples	Ratio (%)	Samples	Ratio (%)
*Aglenchus*	8	4.37	9	6.25	5	7.04	3	11.11
*Antarctylus*	-	-	2	1.39	1	1.41	1	3.70
*Aphelenchoides*	7	3.83	3	2.08	7	9.86	2	7.41
*Aphelenchus*	5	2.73	13	9.03	6	8.45	1	3.70
*Basiria*	7	3.83	1	0.69	-	-	1	3.70
*Boleodorus*	6	3.28	10	6.94	1	1.41	-	-
*Bursaphelenchus*	-	-	1	0.69	-	-	-	-
*Carphodrus*	-	-	1	0.69	-	-	-	-
*Cephalenchus*	-	-	2	1.39	-	-	1	3.70
*Coslenchus*	2	1.09	3	2.08	1	1.41	-	-
*Criconemella*	3	1.64	1	0.69	-	-	-	-
*Ditylenchus*	5	2.73	2	1.39	4	5.63	2	7.41
Dorylaimidae	7	3.83	2	1.39	4	5.63	-	-
Dorylaimoidea	128	69.95	71	49.31	30	42.25	17	62.96
*Ecphyadophoroides*	1	0.55	-	-	-	-	-	-
*Filenchus*	49	26.78	32	22.22	12	16.90	7	25.93
*Gracilacus*	-	-	-	-	1	1.41	-	-
*Helicotylenchus*	17	9.28	38	26.39	28	39.44	9	33.33
*Hemicriconemoides*	6	3.28	-	-	-	-	1	3.70
*Hemicycliophora*	-	-	-	-	1	1.41	-	-
*Heterodera*	-	-	1	0.69	-	-	-	-
*Hirschmanniella*	145	79.23	13	9.03	5	7.04	1	3.70
Hoplolaimidae	-	-	2	1.39	1	1.41	1	3.70
*Hoplolaimus*	2	1.09	1	0.69	7	9.86	2	7.41
Longidoridae	6	3.28	8	5.56	2	2.82	1	3.70
*Longidorus*	1	0.55	2	1.39	3	4.23	-	-
*Malenchus*	4	2.19	5	3.47	1	1.41	-	-
*Meloidogyne*	49	26.78	5	3.47	-	-	3	11.11
*Miculenchus*	4	2.19	5	3.47	1	1.41	1	3.70
*Neopsilenchus*	-	-	2	1.39	2	2.82	-	-
*Paraphelenchus*	4	2.19	5	3.47	-	-	-	-
*Pararotylenchus*	-	-	5	3.47	9	12.68	2	7.41
*Paratrichodorus*	-	-	1	0.69	2	2.82	-	-
*Paratylenchus*	-	-	-	-	-	-	1	3.70
*Pratylenchoides*	-	-	-	-	1	1.41	-	-
*Pratylenchus*	6	3.28	65	45.14	25	35.21	7	25.93
*Psilenchus*	1	0.55	4	2.78	-	-	1	3.70
*Rhabdotylenchus*	1	0.55	1	0.69	1	1.41	-	-
*Rotylenchulus*	1	0.55	2	1.39	1	1.41	-	-
*Rotylenchus*	2	1.09	17	11.81	7	9.86	1	3.70
*Scutellonema*	-	-	1	0.69	-	-	-	-
*Trichodorus*	-	-	2	1.39	6	8.45	-	-
*Triplonchida*	-	-	4	2.78	-	-	-	-
Tylenchidae	1	0.55	-	-	-	-	-	-
*Tylenchorhynchus*	63	34.43	47	32.64	50	70.42	8	29.63
*Tylenchus*	1	0.55	3	2.08	-	-	1	3.70
*Xiphidorus*	1	0.55	-	-	-	-	-	-
*Zygotylenchus*	-	-	1	0.69	-	-	-	-

**Table 2 life-11-01177-t002:** Proportion (%) of main crops with different soil textures in this survey.

Soil Texture	Rice	Maize	Sugarcane	Soybean
Loam	16.67	51.39	26.39	5.55
Clay	90.59	4.71	4.70	0

**Table 3 life-11-01177-t003:** PPNs and crops at different altitudes.

Altitude (m)	Number of Soil Samples	Number of Family/Genus	Main Crop
<50	47	25	Rice
50–100	126	40	Rice, Maize
100–150	121	28	Rice
150–200	67	31	Maize
>200	60	30	Rice

## Data Availability

Data for this study can be made available with reasonable request to the authors.

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
