# Peer review of "Systematic Investigation of Plant-Parasitic Nematodes Associated with Main Subtropical Crops in Guangxi Province, China"

_life, 2021, doi:10.3390/life11111177_

Round 1
Reviewer 1 Report
The authors conducted a survey on nematodes of main crops in a state of China, but there is no mention of the statistical design for sampling nor is there information on number of subsamples per sampled plot/field. The authors also mention that root saples were collected with no in formation on how many and from which crop nor on how nematodes were extracted from the roots. However, the main flaw of this manuscript is that the nematodes were identified at only genus level and not at species level at least for the most important and abundant nematode populations. It is well known that most nematode genera include many species not all equally damaging to crop plants. In the manuscript the authors just mention heterodera zeae and H. glycines without stating on the base of what these two species were identified. Nor is there information on the size of the nematode population densities to figure the extent of damage the nematodes might be causing to the sampled crops. Therefore, the manuscript do not contains information of interest for an international readership. Nevertheless, the results may have interest at local level and, therefore, I suggest the authors to publish them on a local journa.
Author Response
We had revised the manuscript according to the comments.
Reviewer 2 Report
The authors of the manuscript Systematic investigation of plant-parasitic nematodes associated with main subtropical crops in China investigated the distribution of the most abundant plant-parasitic nematodes (ppn) on the main crops in Guangxi Province in south-western China. The findings were related to host crops, soil type, soil moisture, and altitude. This systematic study of ppn distribution is original and of interest for plant pathologists as well as ecologists. The topic is within the scope of the journal.
Although there is nothing intrinsically wrong with the title, the authors might want to narrow it down to the region or province of their investigation (e.g., Guangxi Province, China).
The abstract summarizes the research and results nicely.
The introduction gives a good overview of similar research worldwide and ties in relevant and up-to-date literature citations. It might be helpful to point out that the scope of the investigation did not allow a higher resolution than identification to the genus level.
The M&M are sufficient as the techniques are well known and relatively straightforward.
The results are appropriately listed in tables and figures. The discussion and conclusions are appropriate and supported with sufficient citations.
In the following I have listed some minor suggestions and corrections.
Line
2 [typically hyphenated; throughout the manuscript] ...plant-parasitic...
3 [change word order] ...with main subtropical crops...
14 [comma]... abundant, accounting for
15 [spelling out number at beginning of sentence] Forty...
24 [comma splice; divide into two sentences] ...with different altitudes. The diversity of...
25 [comma] ...PPNs, and provide...
31 [add preposition] ...resulting in...
43 [missing letter]...parasitize...
49 [singular] South Africa has...
58 [change quantifier and add comma] ...fewer species, but a higher...
59 [split sentence] ...in sand. The concentrations...
61 [split sentence]...layer. However,...
62 [typo] ...pointed...
80 [add "a" and delete "by"] ... a 100 mL soil sample by using Cobb's...
86 [replace comma with semicolon] ... 10-12 h; all ...
96 [singular] ... vary...
103 [split sentence] ... Excel 2010. Figures were ...
113 [replace word]...the most abundant with...
149 [split sentence] ... high frequency. In contrast, Hirschmanniella,...
169 [split sentence] ... different altitudes. In contrast, some...
172 [typo] ... Helicotylenchus crenacauda...
173 [comma] ...Ditylenchus, and ...
174 [insert] ...were detected, with the frequencies...
175 [change word and semicolon] ...40 nematode genera; the highest ...
203 [add "a"] ... a total of...
209 [capitalize] ... in Southern China.
210 [split sentence] ... maize in Southern China. The results showed...
211 [capitalize] ... to Northern China, except...
215 [spell out the number; see line 15] Thirty genera...
222 [split sentence] ... in Guangxi. In contrast, Pratylenchus...
248 [change word] ... at 50–200 m, whereas fewer genera were distributed at...
250 [change phrase] On account of samples, Dorylaimoidea ...
254 [change wording] ... grapevine, an essential crop...
258 [unclear; suggestion] ...its importance as a plant pathogen is unknown.
260 [spell out number; see 1ine 15] ...Forty-eight ...
272 [eliminate last period]
Author Response
Response to reviewer 2
- Although there is nothing intrinsically wrong with the title, the authors might want to narrow it down to the region or province of their investigation (e.g., Guangxi Province, China).
Answer: We put ‘Guangxi Province’ in title
In the following I have listed some minor suggestions and corrections.
Line
2 [typically hyphenated; throughout the manuscript] ...plant-parasitic...
Answer: we checked throughout the manuscript and revised it.
3 [change word order] ...with main subtropical crops...
Answer: we changed it according to comments.
14 [comma]... abundant, accounting for
Answer: we revised it.
15 [spelling out number at beginning of sentence] Forty...
Answer: we changed it to ‘Forty’.
24 [comma splice; divide into two sentences] ...with different altitudes. The diversity of...
Answer: we divided it into two sentences.
25 [comma] ...PPNs, and provide...
Answer: we added comma.
31 [add preposition] ...resulting in...
Answer: Yes, we did it
43 [missing letter]...parasitize...
Answer: we corrected it.
49 [singular] has...
Answer: we changed it.
58 [change quantifier and add comma] ...fewer species, but a higher...
Answer: Yes, we revised it according to comments.
59 [split sentence] ...in sand. The concentrations...
Answer: we split this sentence.
61 [split sentence]...layer. However,...
Answer: we split it.
62 [typo] ...pointed...
Answer: we corrected it.
80 [add "a" and delete "by"] ... a 100 mL soil sample by using Cobb's...
Answer: we revised it according to comments.
86 [replace comma with semicolon] ... 10-12 h; all ...
Answer: we changed it.
96 [singular] ... vary...
Answer: we changed varies to vary.
103 [split sentence] ... Excel 2010. Figures were ...
Answer: we split the sentence.
113 [replace word]...the most abundant with...
Answer: we replaced ‘popular’ with ‘abundant’.
149 [split sentence] ... high frequency. In contrast, Hirschmanniella,...
Answer: we split this sentence according to comments.
169 [split sentence] ... different altitudes. In contrast, some...
Answer: we split it.
172 [typo] ... Helicotylenchus crenacauda...
Answer: we corrected it.
173 [comma] ...Ditylenchus, and ...
Answer: we put comma after Ditylenchus.
174 [insert] ...were detected, with the frequencies...
Answer: ‘were’ and ‘with’ were inserted .
175 [change word and semicolon] ...40 nematode genera; the highest ...
Answer: we changed ‘40 kinds of nematode,’ to ‘40 nematodes genera;’.
203 [add "a"] ... a total of...
Answer: we added ‘a’.
209 [capitalize] ... in Southern China.
Answer: we revised it.
210 [split sentence] ... maize in Southern China. The results showed...
Answer: We separated it with a full stop.
211 [capitalize] ... to Northern China, except...
Answer: we did it.
215 [spell out the number; see line 15] Thirty genera...
Answer: we spell out the number with ‘Thirty’.
222 [split sentence] ... in Guangxi. In contrast, Pratylenchus...
Answer: we split the sentence.
248 [change word] ... at 50–200 m, whereas fewer genera were distributed at...
Answer: we changed it.
250 [change phrase] On account of samples, Dorylaimoidea ...
Answer: we changed it to ‘On account of samples’.
254 [change wording] ... grapevine, an essential crop...
Answer: we changed ‘important’ with ‘essential’.
258 [unclear; suggestion] ...its importance as a plant pathogen is unknown.
Answer: we revised it according to comments.
260 [spell out number; see 1ine 15] ...Forty-eight ...
Answer: we spell out number with English.
272 [eliminate last period]
Answer: we deleted the last stop.

Reviewer 3 Report
A well conducted survey which gives useful information on the most widespread,abundant and dangerous genera of plant parasitic nematodes in relation to each crops in the localities investigated
Author Response
Page1 Line 42-46: We added ‘In China, Meloidogyne graminicola, Aphelenchoide besseyi, Hirschmanniella oryzae, Ditylenchus angustus and Heterodera elachista are the nematode species causing major damage to rice and other gramineous crops [10].’
Page2 Line 52-54: We added ‘In China, Tylenchorhychus spp., Helicotylenchus spp., Meloidogyne spp., and Pratylenchus spp. seriously damaged sugarcane [13].’
We revised the manuscript in Page3 Line 131-132 and Line 134-135.
We revised the manuscript in Page6 Line 220-221, Page 7 Line 233 and Page 8 Line 312-313.

Reviewer 4 Report
The manuscript on the plant parasitic nematodes (PPN) in Guangxi province appears well constructed. Here are a few concerns I have:
- It lacks the scientific depth: the authors cited PPN on rice, rice is such an important crop, you need to have the nematode to species level. There are numerous publications of PPN on rice at species level. Rice is the most important crop for Guangxi. I suggest authors go back to the nematodes (I hope you have kept) identified to species level, or seek assistant elsewhere. As for other crops, I let editors to decide if species level is necessary.
- In the introduction, the authors cited more papers outside of China than those inside China. I am sure there are plenty of papers on PPN in China, so add more from China especially from the neighboring provinces which have similar crop profile and climate conditions.
- In the method section, the authors seem to suggest that cysts were found, but in the result section, no cyst nematodes (Heteroderidae) were listed.
- Some attention needed on the grammar.
- The table should list the genus names in alphabetical order
Author Response
- It lacks the scientific depth: the authors cited PPN on rice, rice is such an important crop, you need to have the nematode to species level. There are numerous publications of PPN on rice at species level. Rice is the most important crop for Guangxi. I suggest authors go back to the nematodes (I hope you have kept) identified to species level, or seek assistant elsewhere. As for other crops, I let editors to decide if species level is necessary.
In fact, for important PPN, the identification of species level is our another important work, we identified the cyst nematode on maize published in journal of ‘plant disease (2017, 101(7):1330. doi:10.1094/PDIS-01-17-0146-PDN)’, and root knot nematode on rice published in journal of ‘The Plant Pathology Journal’. We discussed them in discussion section.
- In the introduction, the authors cited more papers outside of China than those inside China. I am sure there are plenty of papers on PPN in China, so add more from China especially from the neighboring provinces which have similar crop profile and climate conditions.
Actually, there are some PPN reported in China, the genus similar to the report of outside China. For example, In China, Tylenchorhychus spp., Helicotylenchus spp., Meloidogyne spp., and Pratylenchus spp. seriously damaged sugarcane (Wei et al., 2012).
We had put this citation in the manuscrtipt.
- In the method section, the authors seem to suggest that cysts were found, but in the result section, no cyst nematodes (Heteroderidae) were listed.
We have this cyst nematode in Table 1 listed (Heterodera, on maize, Line 21 in table 1) in the manuscript.
- Some attention needed on the grammar.
We had changed it carefully.
- The table should list the genus names in alphabetical order
We had revised Table 1, listed the genus names in alphabetical order).

Round 2
Reviewer 1 Report
The authors, answering to my comments, say done. Actually, apart from minor changes, the manuscript as remained the same. In particular the main flaw, lack of identification to species level of the main nematodes, still remain and, therefore, I confirm my previous comment.
Author Response
The authors conducted a survey on nematodes of main crops in a state of China, but there is no mention of the statistical design for sampling nor is there information on number of subsamples per sampled plot/field.
We sampled 5–10 cores around plant were collected with a soil auger in a Z-pattern in each plot. In fact, 3 subsamples per field. However, we just identified the nematode in genus level and did not record the population of the nematode, only calculated the frequency of the nematode in total sample. The purpose is how many genera of PPN on different corp.
The authors also mention that root saples were collected with no information on how many and from which crop nor on how nematodes were extracted from the roots.
We have the longitude and latitude of the sampling field, if necessary we can provide it.

Reviewer 4 Report
- In the Xie et al. (2017) paper, if Guangxi was included, you need to cite specifically the important or common parasitic nematodes in the province, if not cite those in the provinces close to Guangxi.
- Most of the information in the discussion should be in the introduction section.
- Rice is a such important crop, it is important to identify the most important one to species level.
